 **eLIFE**

# Distinct cortical codes and temporal dynamics for conscious and unconscious percepts

Moti Salti[1,2,3,4]*, Simo Monto[1,2,5], Lucie Charles[1,2,6], Jean-Remi King[1,2], Lauri Parkkonen[1,2,5], Stanislas Dehaene[1,2,7,8]

[1]Cognitive Neuroimaging Unit, Institut National de la Santé et de la Recherche Médicale, Gif sur Yvette, France; [2]Neurospin center, Institut d'Imagerie Biomédicale, Gif sur Yvette, France; [3]Department of Brain and Cognitive Sciences, Ben-Gurion University of the Negev, Beer-Sheva, Israel; [4]Zlotowski Center for Neuroscience, Ben-Gurion University of the Negev, Beer-Sheva, Israel; [5]Department of Neuroscience and Biomedical Engineering, Aalto University School of Science, Espoo, Finland; [6]Attention and Cognitive Control Lab Department of Experimental Psychology, University of Oxford, Oxford, United Kingdom; [7]Collège de France, Paris, France; [8]University of Paris-Sud, Orsay, France

**Abstract** The neural correlates of consciousness are typically sought by comparing the overall brain responses to perceived and unperceived stimuli. However, this comparison may be contaminated by non-specific attention, alerting, performance, and reporting confounds. Here, we pursue a novel approach, tracking the neuronal coding of consciously and unconsciously perceived contents while keeping behavior identical (blindsight). EEG and MEG were recorded while participants reported the spatial location and visibility of a briefly presented target. Multivariate pattern analysis demonstrated that considerable information about spatial location traverses the cortex on blindsight trials, but that starting ≈270 ms post-onset, information unique to consciously perceived stimuli, emerges in superior parietal and superior frontal regions. Conscious access appears characterized by the entry of the perceived stimulus into a series of additional brain processes, each restricted in time, while the failure of conscious access results in the breaking of this chain and a subsequent slow decay of the lingering unconscious activity.

*For correspondence: motisalti@gmail.com

**Competing interests:** The authors declare that no competing interests exist.

## Introduction

The scientific investigation of consciousness divides into two major branches (*Koch, 2004*; *Dehaene et al., 2014*). The first branch, which studies the 'state of consciousness', focuses on general vigilance or a person's ability to perceive, interact, and communicate with the environment, by examining the regulation of sleep and waking, and their pathological disruption by coma, epilepsy, and sleep disorders (*Laureys et al., 2004*). The second branch of inquiry, which studies 'access to consciousness', focuses on the processes that make a specific content subjectively experienced—what differences in brain activity distinguish conscious vs unattended or subliminal stimuli (*Dehaene et al., 2006*). To a large extent, both endeavors rely on characterization of the neural activity that underlies the conscious state or conscious perception, and they are traditionally approached in a similar manner (*Dehaene and Changeux, 2011*). In order to examine the neuronal processes that underlie conscious states, the neuronal activity associated with one state of consciousness (e.g., sleep) is compared to another (e.g., awake). Similarly, in order to extract the neural correlates of conscious perception, threshold-level stimuli are presented to the subject such that they are sometimes consciously

**eLife digest** Our senses constantly receive information from the world around us, but we consciously perceive only a small portion of it. Nonetheless, even stimuli that are not consciously perceived are registered in our brain and influence our behavior. This is known as unconscious perception.

Researchers disagree about how brain activity differs during conscious and unconscious perception. Some think that both consciously and unconsciously perceived objects are processed in the same way in the brain, but that the brain is more active during conscious perception. Others think that different neurons process the information in different types of perception.

Salti et al. have now investigated this issue. While recording participants' brain activity, a line was briefly presented in one of eight different possible locations on a screen. The line was masked so it would be consciously perceived in roughly half of the presentations. Participants had to report the location of the line and then say whether they had seen it or had merely guessed its location. Even when they reported that they were guessing, participants identified the location of the line better than by chance, indicating unconscious perception on 'guess' trials. This enabled Salti et al. to compare how the brain encodes consciously perceived and unconsciously perceived stimuli.

Unlike previous studies in which the brain activity associated with 'seen' and 'unseen' stimuli was compared, Salti et al. used a different approach to extract the neural activity underlying consciousness. A classifying algorithm was trained on a subset of the data to recognize from the recorded brain activity where on the screen a line had appeared. Applying this algorithm to the remaining data revealed the dynamics of stimulus encoding. Consciously and unconsciously perceived stimuli are encoded by the same neural responses for about a quater of a second. From this point on, consciously perceived stimuli benefit from a series of additional brain processes, each restricted in time. For unconsciously perceived stimuli, this chain of processing breaks and a slow decay of encoding is observed.

Salti et al., therefore, conclude that conscious perception is represented differently to unconscious perception in the brain and produces more extensive and structured brain activity. Future work will focus on understanding these differences in neural coding and their contribution to the interplay between conscious and unconscious perception.

perceived and sometimes not. Trials are then sorted according to their subjective perception ('Seen' or 'Unseen'), and the neurophysiological signatures associated with these categories are contrasted to extract the correlates of consciousness perception.

Such paradigms have yielded convergent findings (*Koch, 2004*; *Dehaene and Changeux, 2011*; *Dehaene et al., 2014*), but they have also been criticized on several grounds (*Aru et al., 2012*). First, differences in neural activity may stem from physical differences in the stimuli. To control this, many experiments now employ identical stimuli and rely on participants' subjective reports to differentiate seen and unseen conditions. Even then, stimuli also differ in their depth of processing: performance is typically much higher for conscious trials (*Lau and Passingham, 2006*), and several operations may only be feasible in the conscious state (*Sackur and Dehaene, 2009*; *De Lange et al., 2011*). The main effect of seen vs unseen trials may, therefore, reveal processing differences unrelated to the actual cerebral encoding of a conscious or unconscious stimulus, and reflecting solely, the operations that precede follow or even coincide with conscious access (*Aru et al., 2012*; *Pitts et al., 2012*; *Frässle et al., 2014*; *Pitts et al., 2014*).

Another limitation is that these paradigms do not isolate the brain mechanisms underlying the conscious representation of a specific content (*Haynes, 2009*). The broad contrast between brain activity on 'seen' vs 'unseen' trials may include generic processes of attention, alerting, or reporting that should be carefully kept distinct from the more limited set of processes that represent the current mental content. For instance, the late P3 component of event-related potentials, which is a frequent signature of conscious perception (*Dehaene and Changeux, 2011*), has been proposed to reflect either a non-specific alerting effect arising from noradrenergic neurons of the locus coeruleus (*Nieuwenhuis et al., 2005*) or the activation of generic reporting processes unnecessary to conscious perception itself (*Pitts et al., 2012, 2014*). To resolve this issue, it is necessary to identify which patterns of brain activity,

unique to conscious trials, faithfully encode the contents of subjective experience and to separate them from other non-specific brain responses (*Haynes, 2009*).

The presence or absence of additional information is a crucial feature that separates major theories of conscious access. Some theories propose that subjective experience emerges from a generic pattern of brain activity without any stimulus-specific change (*Lau and Rosenthal, 2011*). According to these models, the cognitive representation of a stimulus is built during unconscious perception, and the additional activity that gives rise to consciousness tags it as 'perceived' but adds no perceptual or coding benefit. Conversely, other models (*Dehaene et al., 2003*, *2006*) suggest that conscious perception relies on a massive amplification and cortical broadcasting of stimulus-specific information. The Global Neuronal Workspace (GNW) model asserts that perceptual stages may unfold identically on conscious and non-conscious trials, but that, when a stimulus gains access to consciousness, stimulus-specific information is amplified and re-encoded in additional areas, including a prefrontal–parietal network. This 'workspace' maintains the information online and dispatches it to additional processors, thus making the stimulus reportable. This view predicts that, ceteris paribus, additional stimulus information should be present on conscious trials. Finally, theories that associate conscious perception with posterior visual loops (*Lamme, 2006*) or with the integration of information into a coherent whole (*Oizumi et al., 2014*) are agnostic with respect to this issue: although greater reverberation or integration may cause an amplification of neuronal activity, (*Fahrenfort et al., 2012*) it could also, on the contrary, lead to diminished or sparser activity due to the greater predictability of the distributed brain signals being integrated (*Friston, 2005*).

In summary, for both empirical and theoretical reasons, it is essential to study the temporal propagation of conscious vs unconscious information in the brain and to probe the presence of additional stimulus-specific cortical codes on seen trials relative to unseen trials, while excluding any difference in perception and behavior. To fulfill this demanding agenda, we used time-resolved MEG and EEG recordings to track the internal representation of a flashed stimulus under visible and invisible conditions. This strategy was made possible by advances in multivariate decoding, which can detect stimulus-specific information in brain activity, and evaluate its localization (*Kamitani and Tong, 2005*; *Haynes and Rees, 2006*; *Haynes, 2009*) and its time course (*King and Dehaene, 2014*). We chose visual location as the decoded parameter of the stimulus, because multiple macroscopic retinotopic maps are present throughout the human brain, including the frontal lobe (*Sereno et al., 1995*; *Hagler and Sereno, 2006*), and therefore, the full cortical processing stream of this parameter should be more easily decodable than other more microscopic parameters, such as object identity. To further address the concern of differences in processing depth, we placed ourselves under 'blindsight' conditions by asking subjects to perform a forced-choice location task in which performance was excellent even on subjectively invisible trials. Selecting only the correct trials allowed us to compare visible and invisible trials with identical stimuli and responses (*Lamy et al., 2009*).

## Results

We recorded Magnetoencephalography (MEG) and Electroencephalography (EEG) while presenting the subjects with a tilted line segment at one of eight possible locations arranged circularly around the fixation point. The target was followed by a mask whose contrast was adjusted individually for each participant to ensure conscious perception on roughly half of the trials. In each trial, participants provided three separate behavioral responses: 1) an immediate forced-choice localization of the target, 2) whether their first response was correct or not, and 3) whether they saw the target or not (see *Figure 1A* and 'Materials and methods'). Participants were instructed to report the target as unseen only if they had no perception at all regarding its location. If they did not see the target they were asked to guess its location (response 1).

Behaviorally, subjects reported seeing 62.4 ± 3.62% (mean ± sem) of presented targets, and the accuracy (i.e., reporting the correct location) in these trials was 86.6 ± 2.54%. Performance in the unseen trials was 50.7 ± 6.4% correct, the way higher than the chance level of 12.5% ($t(11) = 5.73$, p <0 0.001), indicating blindsight. At all eight stimulated locations, the dominant response was to correctly point to that location, even on unseen trials (see *Figure 1B*). Those 'unseen–correct' trials thus provide a minimal contrast with 'seen–correct' trials. Mean localization Respone Times (RTs) were 812.2 ± 44 ms. Localization RTs for different critical conditions were as follow: 701.4 ± 35 ms for 'Seen–Correct' trials, 878.1 ± 78 ms for 'Unseen–Correct' trials, and 1110 ± 101 ms for 'Unseen-incorrect' trials.

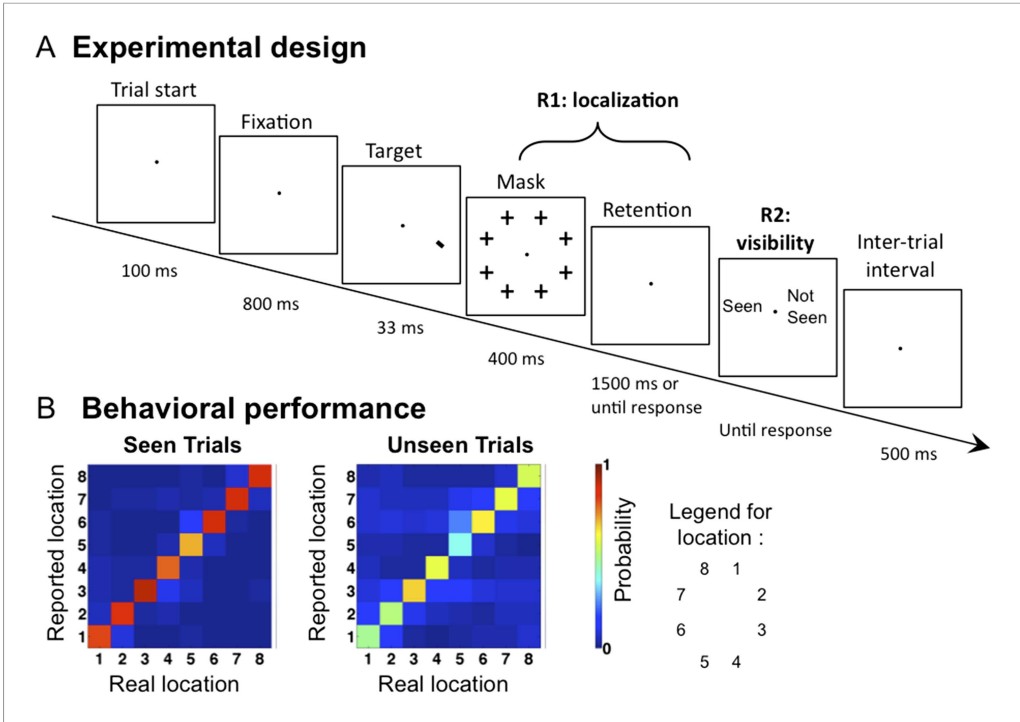

**Figure 1**. Experimental design and behavioral results. (**A**). Sequence of events presented on a trial. Subjects attempted to localize a brief target, which could appear at one of eight locations. Mask contrast was adjusted to ensure ~50% of unseen trials. On 1/9 of trials, the target slide was replaced by a blank slide (target-absent trial). (**B**) Behavioral confusion matrices describing the distribution of responses for each spatial location when target was seen (left matrix) or unseen (right matrix). A strong diagonal on unseen trials indicates blindsight.

## Time course of location decoding

To track the cortical representation of the target location through time, we trained multivariate pattern classifiers separately at every time sample to predict the target location from the recorded brain signals (see 'Materials and methods'). For each time sample, we extracted classifiers' calibrated posterior probabilities (*Platt, 1999*) associated with the eight spatial locations. We first trained the classifiers on all trials, regardless of participants' responses. *Figure 2A* depicts the classifiers' assigned probability for the correct target location as a function of time. Four stages could be distinguished (this division will be used throughout our analyses). In a first stage (0–115 ms post stimulus), classification probability was at chance ($0.1251 \pm 0.0001$; mean $\pm$ sem), t < 1. From 115 to 162 ms, decoding probability rose above chance and peaked sharply at 147 ms post stimuli ($0.141 \pm 0.0028$), t(11) = 5.9, p < 0.0001. Then came a transition period (162–271 ms) with reduced but still significant decoding ($0.137 \pm 0.0023$), t(11) = 5.48, p < 0.0001. Finally, in a fourth stage (271–800 ms), decoding probability rose again and remained above chance for about 500 ms ($0.157 \pm 0.0083$), t(11) = 3.8, p < 0.0001.

## Decoding on seen and unseen trials

We next examined how classifier performance varied with the subject's report and performance (See *Figure 2B*). Our behavioral procedure allowed us to differentiate between two levels of processing in the 'Unseen' category: 'Unseen–Correct' trials where the response contained genuine location information, and 'Unseen-Incorrect' in which it did not. Remarkably, even in the latter trials, where behavioral evidence did not indicate any processing of stimuli, posterior probabilities for decoding spatial information were higher than chance even in the latest time window ($0.143 \pm 0.005$), t(11) = 3.37, p < 0.01. This result is striking for two reasons; first, it allows us to broaden the concept of blindsight as observed in V1 lesioned patients (*Weiskrantz, 1996*)

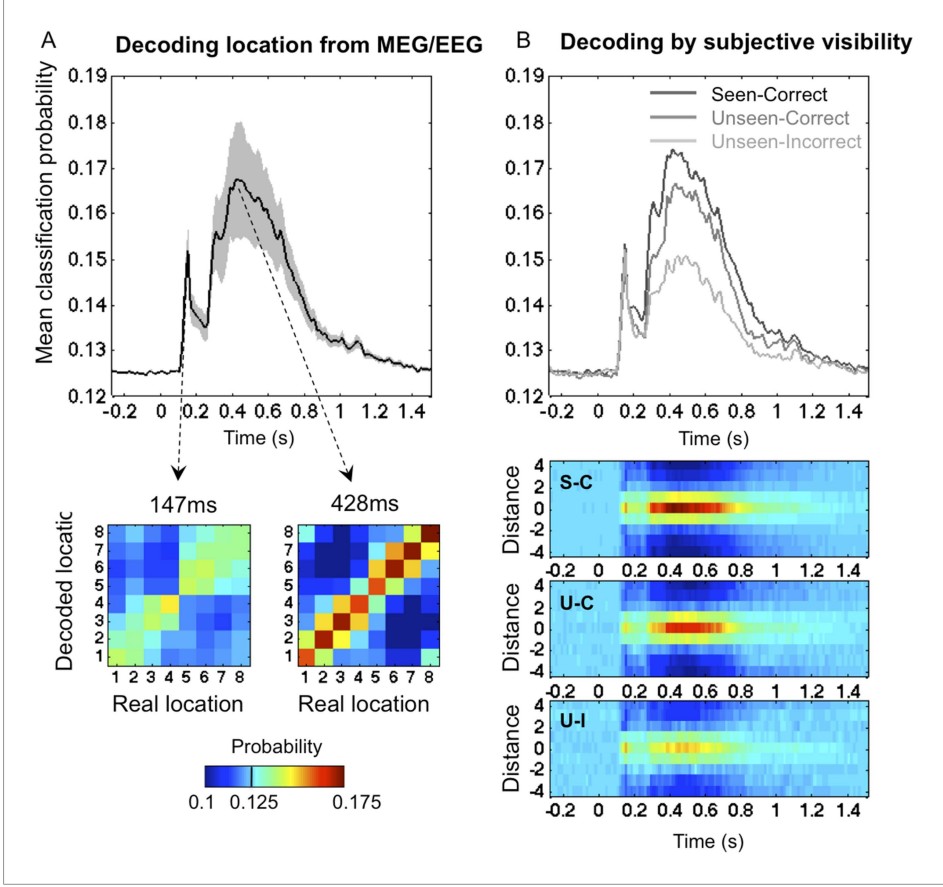

**Figure 2**. Time course of location information. (**A**). Average posterior probability of a correct classification of target location, as a function of time. Chance = 12.5% (1/8). Decoding confusion matrices are shown at the two decoding peaks. (**B**) Same data sorted as a function of subjective visibility (seen/unseen) and objective localization performance (correct/incorrect). The lower part shows the time course of average classifier probability as a function of distance between the decoded and actual target location.

The following figure supplements are available for figure 2:

**Figure supplement 1**. Event-Related Fields and potentials for 'Seen–Correct' vs 'Unseen–Correct'.

**Figure supplement 2**. Event-Related Fields and potentials for 'UnSeen–Correct' vs 'Unseen–InCorrect'.

**Figure supplement 3**. Classifying visibility.

**Figure supplement 4**. Controls for eye movements and motor-based decoding.

and monkeys (*Cowey and Stoerig, 1995*) or induced in normal subjects (*Lamy et al., 2009*). In blindsight, above-chance forced-choice behavior indicates that the observer perceived the stimuli, even though he denies any subjective perception. Here we can see that, even when subjects fail to perform the forced-choice task, their brain still perceived and retained the spatial information. Although unconscious perception is considered as transient and short-lived (*Rossetti, 1998*), here we see that even with the lowest end of perception, spatial information is encoded for up to 800 ms.

Nonetheless, the dynamics of classification probability associated with 'Unseen–Correct' and 'Unseen-Incorrect' were not identical; they shared the same classification probability for the first three stages but diverged in the fourth stage (271–800 ms) where the mean classification probability was higher for 'Unseen–Correct' (0.155 ± 0.009) than for 'Unseen–Incorrect' (0.143 ± 0.005). These results

indicate, unsurprisingly perhaps, that additional information about location is available in the brain on trials in which subjects responded correctly than in trials in which they did not.

The main aim of this paper was to track the temporal dynamics of neuronal encoding of consciously and unconsciously perceived spatial information, that is, the differences between 'Seen–Correct' and 'Unseen–Correct', which were strictly identical in terms of both stimuli and responses. 'Seen–Correct' and 'Unseen–Correct' trials did not differ in classification probability during the first three stages but diverged only during the fourth stage (see *Figure 2B*). Mean classification probability between 272–800 ms was higher for 'Seen–Correct' trials (0.162 ± 0.009) than for 'Unseen–Correct' (0.155 ± 0.009), t(11) = 2.93, p = 0.014. This difference survived correction for correct responses produced by chance (*Lamy et al., 2009*) in the 'Unseen–Correct' condition. Classification was higher for 'Seen–Correct' trials (0.162 ± 0.009) than for 'Unseen–Correct$_{ChanceCorrected}$' (0.157 ± 0.009), t(11) = 2.93, p = 0.029 (one-tailed).

## Relation to previous findings from event-related responses

Our fourth time window, where significantly improved location decoding was observed on seen compared to unseen trials, corresponds to the timing of N2 and especially P3 event-related potentials, which several past studies using the traditional contrast of 'Seen' vs 'Unseen' unseen trials have associated to conscious perception (see *Dehaene and Changeux, 2011* for review). To examine whether our study replicated those findings, we compared the event related potentials (ERP) - event related fields (ERF) signatures of our critical conditions (see *Figure 2—figure supplement 1* and *Figure 2—figure supplement 2*). We tracked the components by computing Global Field Power and applied cluster analysis across channels. While the 'Unseen–Correct' vs 'Unseen-Incorrect' did not yield any significant difference in evoked responses, the 'Seen–Correct' vs 'Unseen–Correct' did: MEG magnetometers showed different activities in the N2 time window, while EEG electrodes showed a difference in the P3 time window, with the appropriate topography (*Figure 2—figure supplement 1*). Thus, these late events again appear as correlates of conscious perception. Unsurprisingly, ERP-ERF analysis revealed a slightly different temporal pattern than the dynamics revealed with the multivariate classification, because the former corresponds to a search for generic brain states allowing or not allowing content to access consciousness, while the latter corresponds to a search for brain states encoding a specific consciously perceived content.

Training a classifier to use those event-related responses in order to separate 'Seen–Correct' vs 'Unseen–Correct' trials yielded a modest but significant classification success, again based on the late part of the epoch (see *Figure 2—figure supplement 3*). Our results suggest that, in the present experiment, M/EEG recordings are dominated by content-specific information about location, both on seen and on unseen trials, while it is more difficult to train a generic decoder for seen/unseen trials that cuts across all such contents. It also should be noted that this experiment was not optimally designed for such an analysis. In order to restrict classifier to using attributes that are related exclusively to visibility and not to stimuli location, we equated the number of trials representing specific stimulus location in the 'Seen–Correct' and the 'Unseen–Correct' location. This reduced the number of trials on which decoder was trained on and affected classification results.

## Controls for hand and eye movements

Because there was a fixed mapping between stimulus location and motor responses (made using four fingers of each hand), the decoding of location could conceptually be based on conscious motor codes. To evaluate this possibility, we performed a control analysis in which we applied the decoder to target absent trials in which no stimulus appeared, but subjects still had to produce a response. Classification of subjects' responses in these trials was at chance, indicating that the previous decoder focused strictly on stimulus-based location information (see *Figure 2—figure supplement 4*). Although it remains possible that conscious response planning modulates the late stages of location processing, several aspects of our experiments argue against a contribution of motor codes to our results. In terms of experimental design, our main analyses focused on trials in which the decoder was trained to extract the objective location of the target on all trials, errors included. Thus, the above-chance extraction of the objective target location on 'Unseen-Incorrect' trials is not likely to reflect motor code, nor can the improved classification performance for 'Seen–Correct' over 'Unseen–Correct', since those trials involve exactly the same stimuli and responses.

Another control analysis evaluated the role of microsaccades. Although eye movements were carefully removed with principal component analysis (PCA), it could be suggested that residual ocular activity in the MEG/EEG (MEEG) signal governed the classification of spatial location. Residual eye movement artifacts, if present, should primarily affect the anterior eye channels. To test this possibility, we examined the dynamics of classification on electro-oculograms (EOG) channels alone. Classification revealed a different temporal pattern of classification that was observed with the full MEEG classification: above-chance location decoding started only around 250 ms, remained much lower that with the whole data set (~14%, where chance = 12.5%), and crucially, did not discriminate between the seen and unseen trials (see *Figure 2—figure supplement 4*), suggesting that eye movements did not play a dominant role in the above decoding results. Source localization (see below) is also incompatible with a single eye movement artifact.

## Asymmetrical generalization of decoding

Our basic analysis indicated that consciously perceived stimuli are coded more reliably than unconscious ones, but the nature of this difference could not be completely appreciated from this analysis. 'Seen–Correct' trials were more numerous than 'Seen-incorrect' trials, and this factor alone could perhaps explain why they were more efficiently decoded. In this case, if conscious and unconscious codes differ, then training specifically on unconscious trials should revert the pattern and yield better decoding on unseen compared to seen trials. The GNW model (*Dehaene et al., 2003*), however, makes a different prediction concerning the asymmetry of decoding on seen and unseen trials. Specifically, the model predicts that all the processing stages present on unseen trials should continue to be observed on seen trials, while the converse should not be true: on seen trials, there should be brain activity encoding the perceived stimulus within the GNW and not present on unseen trials.

We examined these two options by testing for cross-condition generalization: we trained the decoder on a subset of trials (either the unseen–correct trials, or the seen–correct trials) and then tested for generalization, either to left-out trials within the same category or to trials belonging to the other category. One technical difficulty was that, since the experimental conditions were defined by subjects' responses, the number of trials at each location was no longer balanced within each of these conditions, and this imbalance affected the pre-stimulus bias of the 8-category classifier as well as its capacity to generalize. To address this problem, we turned to a simpler binary classifier, which was trained to simply sort the stimuli into left-hemifield vs right-hemifield stimuli. Within each subject, we matched the number of trials in these two classes and also selected equal numbers of seen–correct and unseen–correct trials. *Figure 3* shows that this binary decoder, now operating with a chance level of 50%, still performed above chance with approximately the same time course as the 8-location decoder.

The end result demonstrated the predicted asymmetry. When trained on 'Unseen–Correct' trials, classification generalized identically to 'Seen–Correct' trials, but when trained on 'Seen–Correct' trials, classification performance dropped when tested on 'Unseen–Correct' (*Figure 3*). This effect was apparent in two time windows, the third time window (178–225 ms; average classification probability for 'Seen–Correct' = 0.526 ± 0.0064; for 'Unseen–Correct' = 0.515 ± 0.005; t(11) = 2.8, p = 0.017), and the fourth late time window (272–800 ms; 'Seen–Correct' = 0.575 ± 0.017; 'Unseen–Correct' = 0.547 ± 0.016; t(11) = 2.44, p = 0.032). These analyses indicate that seen–correct trials contained the same decodable stimulus information as unseen–correct trials, plus additional information unique to conscious trials.

## Decoding cortical sources

The GNW model predicts that the additional information associated with conscious representation should be jointly encoded in a network of distributed prefrontal and parietal regions. In order to assess the contribution of these brain areas to the encoding of consciously perceived contents, we modeled the brain activity in 68 regions of interest covering the whole cortex (See 'Materials and methods') and re-trained our decoders using just the distributed source signals from these regions. The results with eight output classes (*Figure 4—figure supplement 1*) confirmed the existence of two main stages: an early one (~100–200 ms) where location information was essentially confined to occipital, ventral visual, and lateral and mesial parietal regions, and a later one (>250 ms) where it became highly distributed to multiple cortical areas, particularly in prefrontal and anterior cingulate

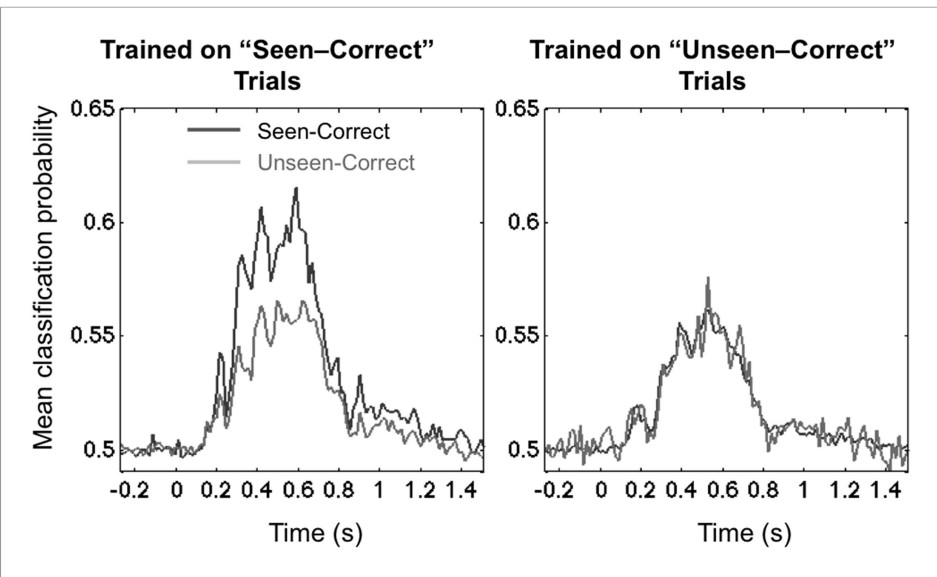

**Figure 3**. Asymmetrical cross-condition generalization. A classifier trained in one condition and then tested on new data either from the same condition or the other condition (e.g. trained on 'Seen–Correct' trials and tested on new 'Seen–Correct' trials and 'Unseen–Correct' trials). To equalize the number of trials, the classifier was trained to discriminate left- vs right-hemifield targets, hence chance = 50%.

cortex (*Figure 4—figure supplement 1*). To avoid an unnecessary increase in the number of statistical tests, we confined the subsequent analyses of seen and unseen trials, to four a priori regions of interest covering all areas of the dorsal visual pathway containing retinotopic maps for location (*Sereno et al., 1995*; *Hagler and Sereno, 2006*): pericalcarine, superior parietal cortex, rostral medial frontal cortex, and superior-lateral frontal cortex. We then repeated the generalization across conditions and procedures. This region-based decoding revealed the activation of successive spatial codes (*Figure 4*): both pericalcarine and superior parietal cortices contained decodable information as early as 115 ms post stimuli ([0.519 $\pm$ 0.005], t(11) = 3.82 p = 0.0028 and [0.516 $\pm$ 0.003], t(11) = 4.52 p < 0.0001, respectively). The two frontal areas revealed a more delayed pattern as the curve associated with superior frontal cortex showed first above-chance decoding only after 194 ms (0.504 $\pm$ 0.001), t(11) = 2.33 p = 0.0396 and rostral medial frontal as late as 365 ms.

Only two of the chosen regions, superior frontal and superior parietal cortices, demonstrated the asymmetry between seen and unseen trials obtained with sensor-based decoding. In both cases, when trained on activity restricted to the superior frontal regions, a complete generalization obtained when trained on 'Unseen–Correct' but when trained on 'Seen–Correct' trials, decoding probability dropped for 'Unseen–Correct' in the fourth time window. Classification probability was significantly higher for 'Seen–Correct' (0.552 $\pm$ 0.017) than for the 'Unseen–Correct' trials (0.541 $\pm$ 0.017), t(11) = 2.35, p = 0.038. In the superior parietal cortex, when trained on 'Unseen–Correct' trials, asymmetry was apparent in all time windows: classifiers generalize completely for 'Seen–Correct' trials (*Figure 4*) but when trained on 'Seen–Correct' trials and tested on 'Unseen–Correct', posterior probabilities dropped. In the second window classification, probability was significantly higher for 'Seen–Correct' (0.534 $\pm$ 0.007) than for the 'Unseen–Correct' trials (0.525 $\pm$ 0.006), t(11) = 2.5, p = 0.026. In the third time window, 'Seen–Correct' decoding probability was higher than 'Unseen–Correct' ([0.537 $\pm$ 0.008] [0.523 $\pm$ 0.01], t(11) = 2.3, p = 0.036). This pattern was kept in the fourth time window, as 'Seen–Correct' decoding probability (0.559 $\pm$ 0.011) was higher than 'Unseen–Correct' decoding probability (0.545 $\pm$ 0.01), t(11) = 3.9, p = 0.002.

These findings go beyond the distinction of conscious and unconscious processing and allow us to demonstrate a representation of stimulus location outside visual areas. Above-chance classification could be achieved in all regions, including superior parietal, rostral middle frontal, and superior frontal

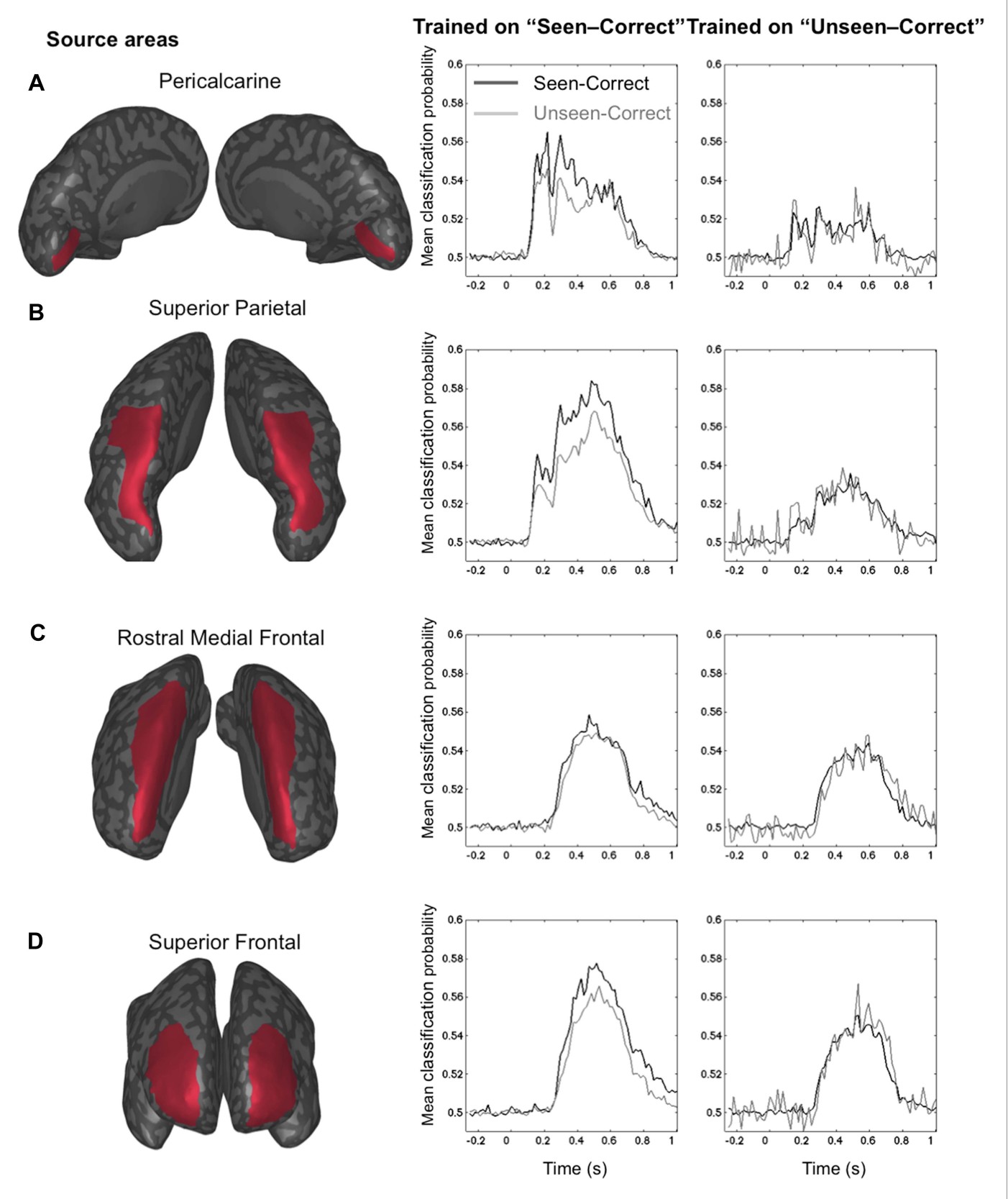

**Figure 4**. Source-based decoding and cross-condition generalization. Classifiers were trained as in *Figure 3*, but using a restricted subset of cortical
*Figure 4. continued on next page*

*Figure 4. Continued*

sources: pericalcarine (**A**), superior parietal (**B**), rostro-medial frontal (**C**), or superior frontal (**D**). Note, how asymmetrical cross-condition generalization (right columns, same format as *Figure 3*) successively arises in visual cortex, then superior parietal, and superior frontal regions.

The following figure supplement is available for figure 4:

**Figure supplement 1**. Time course of location information for the different cortical sources.

regions, supporting the prior finding that these regions contain retinotopic maps (*Sereno et al., 1995*; *Hagler and Sereno, 2006*). Importantly, however, our results show that there is no specific region uniquely dedicated to the encoding of consciously perceived information. Instead, a better encoding on seen trials is obtained in the same overall sectors of the superior parietal cortex and superior frontal cortex that also contain location information on unseen–correct trials (i.e., blindsight trials). Again, the key difference is asymmetrical generalization, indicating the presence of superior encoding of spatial information on seen trials, only in dorsal parietal and frontal regions. This pattern primarily occurs in the late time window (beyond 270 ms), but it is already present at early stages of processing in the superior parietal cortex.

## Generalization across time

According to the GNW model, conscious access corresponds to an amplification and broadcasting of the selected information. Conscious processing is, therefore, predicted to consist in a series of information-processing stages, each associated with the settling of brain activity into a temporary metastable state (stable for a duration of ~100–300 ms), which allows the information to be flexibly transmitted to the appropriate specialized processors (*Dehaene et al., 2014*). The series of processing stages may differ as a function of task demands and may include verbal report (*Frässle et al., 2014*), executive control (*Van Gaal et al., 2010*), or meta-cognitive evaluation (*Charles et al., 2014*). On unconscious trials, the accumulation of incoming activation would fail to attain the critical threshold level needed to trigger such a discrete series of stages ('failed ignition'), and the unconscious information would, therefore, decay over a period of a few hundreds of milliseconds (*Dehaene and Naccache, 2001*).

To evaluate those predictions concerning the dynamics and transient stability of internal codes, we used the temporal generalization method (*King and Dehaene, 2014*). This method consists in probing if a classifier trained at a certain time point $t$ can generalize to other time points, $t'$. If the representation is stable, the classifier should remain efficient even if applied at a different latency. If, however, the information is successively re-encoded in a series of different brain systems, then we should see a failure of generalization beyond a certain temporal duration (i.e., away from the diagonal where $t = t'$) (*King and Dehaene, 2014*). We, therefore, quantified the endurance of conscious and unconscious representations by training and testing classifiers on all pairs of time samples $(t,t')$. To measure a classifier's durability independently of classification efficacy, we defined Classification Endurance (CE), as the number of samples forward and backward in time, for which decoding performance remained above 50% of its level at the original training time $(t = t')$—that is, the decoder's half-life time.

Matrices of temporal generalization (*Figure 5*) indicated that, up to 272 ms, decoding was narrowly restricted to the diagonal for both seen and unseen trials, suggesting a fast-changing chain of perceptual processes (*King and Dehaene, 2014*) consistent with a feedforward propagation of unconscious location information. Beyond this point and up to ~800 ms post stimulus, classifiers generalized to a wider neighborhood of latencies, consistent with the entry of information into evidence accumulation systems with longer time constants. The generalization pattern was examined using an analysis of variance conducted on CE in the 272–800-ms time window with factors of Visibility (Seen/Unseen), Direction (Forward/Backward in time), and Timeframe (5 successive time windows) (see *Figure 5—figure supplement 1* for statistics). All main effects and their interactions were significant. On average, CE was larger for generalization forward than for generalization backward and also increased in time. We were mainly interested in the different generalization patterns of the 'Seen–Correct' and 'Unseen–Correct' trials. Surprisingly, generalization was more extended in time

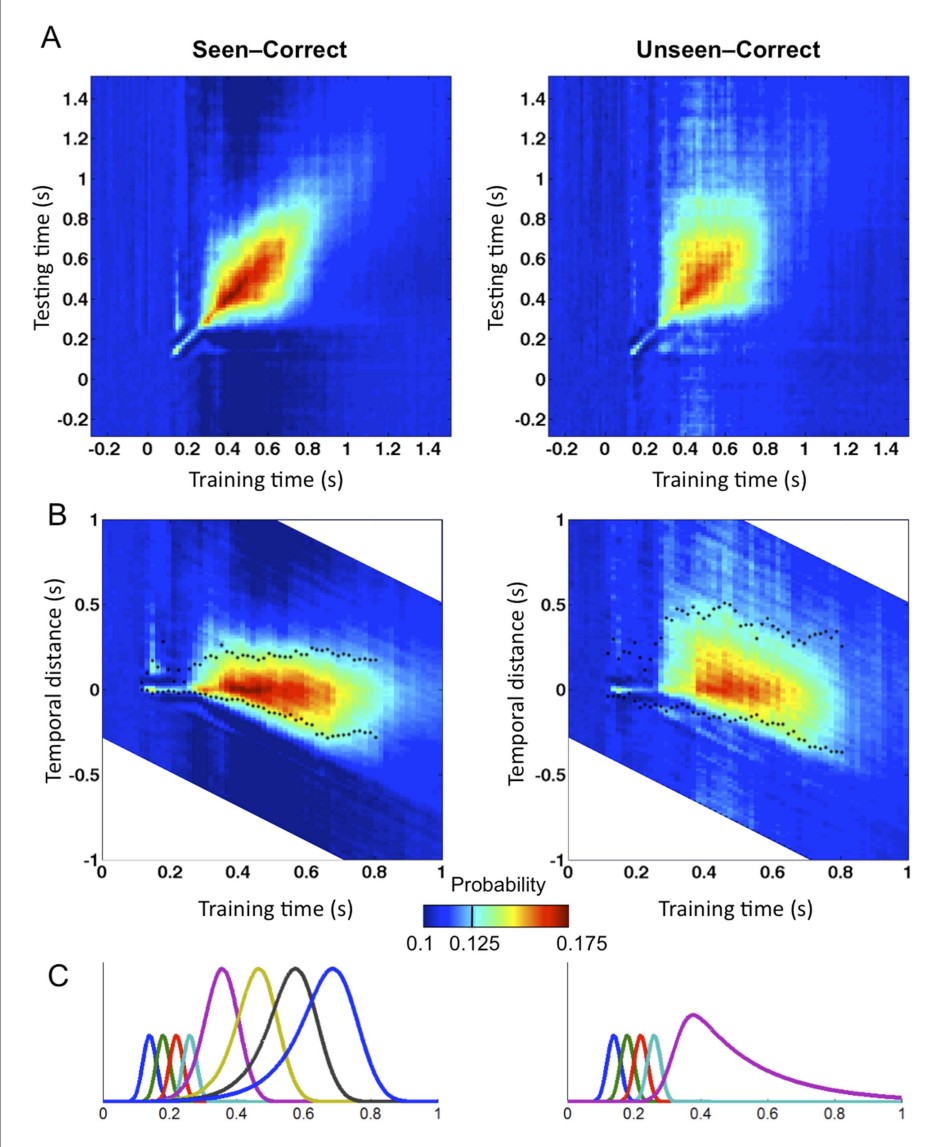

**Figure 5**. Generalization of location decoding over time. 8-location classifiers trained at a specific time were then tested on data from all other time points. (**A**) Average classification probability as a function of testing time for each training time (the diagonal, where testing time = training time, gives the curve for classical decoder performance over time). (**B**) Same information plotted as a function of temporal distance from training time (positive or negative), with asterisks indicating the Classification Endurance (CE) measure. (**C**) Tentative model of a sequence of brain activations, which could yield the observed generalization matrices.

The following figure supplements are available for figure 5:

**Figure supplement 1**. Analysis of variance (ANOVA) on Classification Endurance (CE).

**Figure supplement 2**. ANOVAs on CE with factors of Visibility (Seen–Correct vs Unseen–Correct) and Timeframe (5 levels), separately for forward and backward generalization.

**Figure supplement 3**. Significance Values of the Effect of Timeframe (5 levels) for forward and backward generalization in Seen and unseen trials.

for unseen trials. Forward generalization showed a larger CE in unseen trials compared to seen trials (mean = 382 ms vs 201 ms, $p < 0.005$), and the same pattern was apparent for backward generalization to a lesser extent (mean = 159 ms vs 78 ms, $p < 0.05$) (see *Figure 5—figure supplement 2* for statistics). Furthermore, in the forward direction, these differences interacted with time. Forward CE was constant in seen trials, while for the unseen trials, it was linearly decreasing with time. Backward CE did not differ for seen and unseen trials, but increased steadily as a function of time (see *Figure 5—figure supplement 3* for statistics).

*Figure 5C* shows a possible theoretical interpretation consistent with those results. The high level of decoding observed on seen trials, over a long time window of ~270–800 ms, combined with a low level of off-diagonal temporal generalization beyond a fixed horizon of ~160 ms, implies that a conscious location is successively encoded by a series of about four distinct stages, each terminating relatively sharply after a processing duration of ~160 ms. Conversely, the lower decoding level and longer endurance observed for unconsciously perceived stimuli indicate that they do not enter into the same processing chain. The square pattern of the temporal generalization matrix on unseen trials (*Figure 5A*) suggests that unconscious information lingers in a single representational state, which decays slowly over time, thus yielding a paradoxically longer endurance than on seen trials in the forward direction. All of these findings are consistent with the above predictions as well as with earlier decoding results indicating a chain of additional processing stages unique to conscious trials (*Charles et al., 2014*). They suggest that conscious access can be described as the entry of information into a dynamic routing system (*Sackur and Dehaene, 2009*; *Dehaene et al., 2014*) in which information is stabilized, transferred to a processor, exploited in a determined time, and then transferred again to the next stage. As described in accumulation-of-evidence models of decision making (*Shadlen and Kiani, 2011*), a failure to attain a threshold-level activation at one of these stages cuts the process short, prevents the attainment of a conscious-level representation, and causes the information to linger and decay.

## Discussion

Previous experiments exploring the signatures of conscious access have primarily contrasted the overall brain activity evoked by perceived and unperceived stimuli. However, this comparison may be contaminated by non-specific attention, alerting, performance, and reporting confounds. The quest for brain mechanisms of consciousness implies an isolation of neural populations that encodes the details of subjective experience (*Haynes, 2009*). As an effort in this direction, we applied multivariate decoding techniques in order to track the neural encoding of a conscious or unconscious representation of stimulus location through time. To control over stimulus and performance confounds, we capitalized on the blindsight phenomenon and selectively analyzed a large fraction of trials with identical stimuli and accurate responses, which differed only in subjective reports of seeing or not seeing the stimulus.

The results revealed that (1) the location of a briefly flashed stimulus can be accurately decoded from MEG and EEG signals for up to 800 ms, whether or not the stimulus is seen. This finding is coherent with the existence of multiple spatial maps in occipital, parietal, and frontal cortex (*Sereno et al., 1995*; *Hagler and Sereno, 2006*). (2) Seen and unseen stimuli are initially encoded identically, but after ~270 ms, the information is selectively amplified on 'seen' trials. This observation replicates and extends earlier observations with seen and unseen words, digits and pictures during masking (*Del Cul et al., 2007*; *Fisch et al., 2009*), and the attentional blink (*Sergent et al., 2005*). (3) Asymmetrical generalization indicates that shared spatial codes are active on unconscious and conscious trials, but that conscious trials also contain additional neural codes for stimulus location that are absent or weaker on unconscious trials. Source localization traces those conscious codes to superior parietal and superior frontal cortex. While the present method does not allow us to decide whether the difference is quantitative (the same codes are activated more strongly) or qualitative (additional neural codes are recruited), they are convergent with the prior observation that, of the many maps for space in the cortex, superior parietal and frontal maps are selectively amplified by attention, while earlier retinotopic maps are activated in an automatic manner (*Saygin and Sereno, 2008*). (4) The dynamics of cortical coding, as revealed by temporal generalization matrices (*King and Dehaene, 2014*), also differs on seen and unseen trials: conscious information undergoes a series of representational transformations, such that each decoder fails to generalize beyond ~160 ms, while unconscious information, although weaker, generalizes over a longer period.

The latter finding fits with earlier observations that only conscious stimuli can be passed through a series of discrete information-processing stages (*Sackur and Dehaene, 2009*; *De Lange et al., 2011*; *Charles et al., 2014*). The observed characteristic peak-to-end duration of about 160 ms is also compatible with the finding that conscious percepts tend to be locked to an ongoing theta (~4–7 Hz) or high delta (~1–3 Hz) rhythm (*Melloni et al., 2007*; *Doesburg et al., 2009*; *Nakatani et al., 2014*; *Sitt et al., 2014*), which is also the frequency range of slow positive event-related potentials, such as the P3. Doesburg et al. found that perceptual switching in a binocular rivalry paradigm corresponds to the emergence of synchronized gamma rhythm embedded in theta envelope (*Doesburg et al., 2009*). In an attentional blink paradigm, the synchrony of fast oscillations (beta and gamma) with slow activity (theta and high delta) increased with practice, in parallel to improved stimulus visibility (*Nakatani et al., 2014*). Theta activity was also shown to be higher on visible trials compared to invisible trials in a masking paradigm (*Melloni et al., 2007*) and was successfully used to classify conscious vs vegetative patients (*Sitt et al., 2014*). It thus seems plausible that conscious information is able to enter into a series of computational stages that are betrayed by a series of theta- or delta-like peaks (*Dehaene and Sigman, 2012*). Conversely, unconscious information fails to be sequentially dispatched to the successive steps of a serial task and, therefore, remains blocked within a fixed representational state, where it slowly decays. Behavioral priming studies confirm that unconscious information lingers at detectable levels for hundreds of milliseconds (*Greenwald et al., 1996*) or even seconds (*Soto et al., 2011*).

Our results are incompatible with theories that postulate identical codes for conscious and unconscious information, which would only be distinguished by second-order metacognitive tags (*Lau and Passingham, 2006*). They fit with theories that postulate that a conscious episode is distinguished by an amplification of incoming sensory information, possibly through reverberating loops (*Lamme, 2006*), and its distributed representation in multiple distinct regions including parietal and prefrontal cortices (*Dehaene and Naccache, 2001*; *Dehaene et al., 2006*). These results are highly convergent with intracranial local-field potentials and single-neuron recordings during binocular rivalry and continuous flash suppression (*Sheinberg and Logothetis, 1997*; *Kreiman et al., 2002*; *Panagiotaropoulos et al., 2012*), which indicate that late neural discharges in higher cortical areas reflect which of two rivaling images are subjectively perceived. In the case of object identity, decodable neural activity is found in inferior and anterior temporal cortex (*Sheinberg and Logothetis, 1997*; *Kreiman et al., 2002*) but also in prefrontal cortex (*Panagiotaropoulos et al., 2012*) as reported here. Thus, this region appears at the confluence of conscious perception of object identity and location, making it a primary candidate for the integrated perception of a unified conscious scene or event (*Oizumi et al., 2014*).

While numerous previous studies have compared brain activity evoked by 'seen' and 'unseen' stimuli, the advantage of the present study is to specifically pursue the neural dynamics underlying a specific conscious content (stimulus location). Figuratively speaking, one may say that previous studies have extracted the 'canvas' on which conscious perception is painted, while our approach is aimed at extracting the 'painting' itself. Nevertheless, this approach suffers from several limitations. One of them is the low level of decodability achieved: single-trial EEG and MEG responses have a low signal-to-noise ratio, which limits our inferences and would not afford, for instance, a reliable brain–computer interface. Another limit of our design is that it reduces the rich experience associated with conscious perception to a single feature of a very lean stimulus, presented at threshold, which is the sole focus of attention and which is reported on every trial. Whether the present results would generalize to broader real-life situations, especially in the absence of an overt report is an important question for further research. Some recent experiments suggest that removing the need to report radically reduces the neurophysiological correlates of conscious perception, including the P3b component of event-related potentials, leaving only a posterior mid-latency component (*Pitts et al., 2012*, *2014a*, *2014b*; *Frässle et al., 2014*). However, many of those experiments involved a dual task, which is known to delay and dilute late ERPs (*Sigman and Dehaene, 2008*) and to distort conscious access (*Marti et al., 2010*). It might be preferable to adopt a 'passive attentive' paradigm involving mere attention without report. In such an auditory paradigm, late ERPs including the P3b component were again found to contain decodable information about the conscious percept (*Wacongne et al., 2011*; *King et al., 2013*). A correlation of late neurophysiological activity with perceptibility was also obtained in a variety of neurophysiological studies of human visual perception (*Quiroga et al., 2008*; *Kouider et al., 2013*) or rodent tactile perception (*Manita et al., 2015*). With due caution,

we, therefore, hypothesize that the neurophysiological correlates of conscious perception uncovered here might be generalized to other experimental conditions.

## Materials and methods

### Participants

17 right-handed healthy adults (10 men, 19–30 years of age) with no history of neurological or psychiatric disorders participated the study. This study was ethically approved by CPP IDF 7 under the reference CPP 08 021. All participants reported normal or corrected-to-normal visual acuity. The data of 5 subjects were excluded due to failure in calibration, and these observers reported more than 85% of the target trials as seen (even with maximum masking contrast). The intensity values to which subjects were calibrated ranged from 0 to 227, mean 131.46 (SE = 19.76).

### Stimuli and apparatus

The target stimulus was a line segment subtending a visual angle of 0.33°, tilted by 45° to the left. On target-present trials, the target was randomly presented at one of the eight possible locations. The potential locations were allocated on the outline of an imaginary circle centered at the fixation point with a radius of 2.85° of visual angle. Target locations were separated by radial angles of 45° with an offset of 22.5° from the meridians. The mask comprised crosses, each subtending 0.5° of visual angle, always present at all the eight possible locations. The fixation cross subtended 0.15° of visual angle. On target-absent trials, only the mask was displayed. All stimuli but the mask were black on white background; mask contrast was determined individually for each subject in a calibration phase. Stimuli were generated offline using Matlab (MathWorks, Natick, MA), and their presentation was controlled using the Psychtoolbox package for Matlab (*Brainard, 1997*). Stimuli were presented with a Panasonic PT D7700E-K video projector (refresh rate 60 Hz) to a backprojection screen inside the magnetically shielded at a distance of 1 m from the eyes of the subject. Behavioral responses were collected with two 5-button cylindrical fiber optic response pads ('fORP'; Current Designs Inc., Philadelphia, PA).

### Experimental procedure

At the beginning of each trial, as an alerting signal, the fixation point was magnified twofold for 100 ms and then reverted to its normal size (fixation was present throughout the trial). 800 ms later, the target was presented for 2 screen refresh cycles (~33 ms). The mask followed immediately and remained on the screen for 400 ms. A blank screen was presented for at least 1500 ms or until the first response (retention period). Then, two consecutive images prompting the second and third responses were presented until response. A new trial began 500 ms after the third response (see *Figure 1*).

As noted, three responses were collected on each trial. First, participants were required to produce a speeded forced-choice response to the location of the target. The second response was a second-order response, reporting whether the participants thought that their first response was correct or incorrect. With the third response, participants had to indicate whether they had seen the target or merely guessed its location.

Because of the procedure's high task demand, the experiment was designed in a way that enabled participants to adapt to it. We introduced responses in a gradual way, using a slide that explained the nature of the specific response and a movie that illustrated the corresponding keys. The experiment started with a description of the localization task, which was immediately practiced during an eight-trial practice block. In this block, the mask contrast was set to zero, so targets were completely visible, and participants had to correctly respond with the designated response-pad buttons. A correct response turned the target's color to green and an incorrect response turned it to red.

Next, participants were introduced to the second-order response and performed another training block with three objectives: first, to continue the gradual adaptation to the task; second, to ensure a fast, automatic response; third, to make sure that subjects complied with the task and caught their occasional errors. In the training phase, the target was completely visible once again, and subjects had to report its location and whether they made an error with their first response. Average localization performance (RTs and accuracy) and average second-order report accuracy were calculated every 5 trials. If the participant localized the target correctly in all five trials, or his/hers average RT was

above 800 ms, a slide appeared before the next training block encouraging him/her to respond faster. On the other hand, if the participant localized the target correctly in less than four trials, the slide prompted him/her to respond slower. If participant's second-order report was correct in less than four of the trials, then the slide urged the participant to be more accurate with his/her second-order report. Training ended after 16 blocks or if the average performance during the last 4 blocks had reached specific criteria of localization performance between 85% and 95%, average RT below 800 ms and the second-order report correct in more than 90% of trials.

Following the training, participants were introduced to the subjective visibility response and started a calibration phase. The calibration phase was designed to determine the mask contrast that would yield an approximately equal number of trials in which the target stimulus would be seen or not seen (a 50% detection threshold). A trial was considered as a 'Seen' trial if the participant reported that he/she had seen the target and had correctly localized it, or if he/she reported the target as seen, failed to correctly localize it, but detected the error. We used a modified version of the threshold estimation procedure described by Levitt (*Levitt, 1971*), changing mask contrast according to participant's 'Seen' trials proportion in a block. We manipulated contrast by adjusting the pixel intensity values in a unified fashion. Initial intensity was set to 230 (range 0–255). The number of trials in each block increased as calibration persisted, so the first two blocks comprised four trials, the third block six trials, the forth eight trials, and from the fifth block on, all blocks comprised ten trials. The change in Red Green Blue (RGB) values was determined by the proportion of seen trials in a block, and the number of calibration blocks that participant had already completed. The maximal possible change after a single block was 80. This kind of change was applied when the proportion of seen trials was either 1 or 0. However, this maximal value decreased as the number of blocks participant has completed increased. The stopping rule for the calibration was that RGB change was smaller than 1.5, or subjects had completed 80 trials.

After mask contrast was determined, participant started the experimental phase where the contrast was fixed. The experimental phase was similar to the calibration phase with the following changes: it included 540 trials divided into 6 blocks, 1/9 of these trials were catch trials.

## MEG/EEG acquisition

MEG and EEG data were low-pass filtered at 330 Hz and recorded simultaneously at 1000-Hz sampling rate with a 306-channel Elekta Neuromag MEG system (Elekta Oy, Helsinki, Finland), which comprises 102 triple sensors (each with two orthogonal planar gradiometers and one magnetometer) in a helmet-shaped array. Four head position indicator coils were placed on the scalp of the subject, and their locations were digitized with respect to anatomical landmarks prior to the MEG recording. By briefly energizing the coils, the head position was measured at the beginning of each block. EEG signal was recorded from 60 electrodes that were referenced to the nose. The ground electrode was on the clavicle bones. Horizontal and vertical EOG and electrocardiogram (ECG) were also recorded for offline rejection of eye movements and cardiac artifacts.

## Preprocessing

Acquired data were processed with the MaxFilter software package (Elekta Oy) that implements the Signal Space Separation (SSS) method (*Taulu et al., 2004*) to suppress ambient magnetic interference. Gradiometers and magnetometers with amplitudes continuously exceeding 3000 fT/cm and 3000 fT, respectively, were marked as bad channels and were interpolated by SSS. Eye blinks, eye movements, and cardiac activity were detected on the EOG and ECG channels, and to suppress these artifacts, data were averaged with respect to the onset of each artifact separately, and PCA was used to determine the dominant components of these artifacts in the MEG/EEG signals. One to three components were removed according to visual inspection.

Using Fieldtrip software (*Oostenveld et al., 2011*), continuous data were low-pass filtered at 30 Hz and cut to 2.5-s epochs starting 500 ms before the stimulus onset. Data were downsampled to 64 Hz. EEG data from one subject were omitted due to technical problems. Thus, data from 12 subjects were analyzed, with one subject missing EEG data.

To combine data from MEG magnetometer and planar gradiometers as well as from EEG channels, a normalization procedure was first applied; the baseline standard deviation was estimated for each channel using all the trials in the experiment, and then all the samples were divided by this standard deviation to yield a z-score, which was entered in the classification algorithm.

## Classification

We used a multivariate classification procedure to characterize the temporal dynamics of information processing when the stimulus was perceived consciously vs unconsciously. A distinct classifier was trained for each subject and for each time sample, using the data from all sensors. In all analyses, we employed a linear support vector machine (SVM) algorithm (with cost parameter C = 1) that was complemented with a continuous output method providing for each sample tested the probability of belonging to each of the possible classes (i.e., spatial locations) (*Platt, 1999*). Classification was done with the package libsvm (*Chang and Lin, 2011*). The data were randomly divided into 15 non-overlapping folds; the classifier was trained on fourteenfolds and tested on the one that was left out; this procedure was repeated 15 times for each time sample, so all fifteenfolds were eventually used for testing.

In the first analysis, the classifier was trained on all the data (except of the test data) and eight output classes (locations). The general classification probability for the correct location was then split according to participants' subjective reports of visibility and their performance in the forced-choice task.

The second analysis was aimed at examining cross-condition generalization, that is, whether the features that are used to decode spatial location in one condition of visibility are the same as those used in the other. Since the experimental conditions were defined by subjects' responses, the number of trials at each location was no longer balanced within each of these conditions, and this imbalance affected the pre-stimulus bias of the 8-category classifier as well as its capacity to generalize. To address this problem, we turned to a simpler binary classifier, which was trained to simply sort the stimuli into left-hemifield vs right-hemifield stimuli. Within each subject, we matched the number of trials in these two classes and also selected equal numbers of seen–correct and unseen–correct trials. Similarly to the previous analysis, for each perceptual condition, the data were divided into fifteenfolds; the classifier was trained on 14 and tested on the one left out and additionally on the converse perceptual condition.

A third analysis aimed to find out in which cortical regions the processing of seen and unseen stimuli differed. We tested the roles of each region by running decoding analyses separately on sources confined to this region.

We first parceled the cortex into 68 regions with FreeSurfer (Desikan–Killiany atlas). This parcellation is based on cortical curvature patterns that are estimated from individual MRI surface reconstructions. While these are large regions, the precision of MEG source reconstruction, combined with the requirements of the decoding approach, does not afford a very small focus. Smaller regions generally did not contain enough reconstructed dipoles with distinctive information to support accurate decoding. Neuronal current sources underlying single-trial MEG and EEG signals were estimated using linear minimum-norm estimates with fixed cortical source orientations. A three-layer volume conductor model based on individual head geometry was used in the forward computations (MNE-Suite [*Gramfort et al., 2014*] Matlab toolbox). Noise covariance was estimated from the baseline periods of all accepted trials. We then ran an individual decoding analysis for each of the 68 regions. The estimated current distribution within a ROI at a given latency was fed to the same SVM analyses as for the sensor-level data. The results with eight output classes confirmed the existence of early focal and late distributed location-coding stages (see *Figure 4—figure supplement 1*). To avoid an unnecessary increase in the number of statistical tests, we confined the subsequent analyses of seen and unseen trials, to four a priori regions of interest: pericalcarine, superior parietal cortex, rostral medial frontal cortex, and superior-lateral frontal cortex. In the latter case, the decoder outputs were calibrated class probabilities for 'Seen–Correct' and 'Unseen–Correct' classes after 10-fold cross-validation.

The fourth type of analysis aimed to test the stability over time. As in the first analysis, data from all trials were used to train classifiers at each time point ranging from 200 ms before to 1500 ms after the target and then test generalization to every other time point, as described in (*King and Dehaene, 2014*; *King et al., 2014*).

Two additional control analyses were conducted. First, we used cross-condition generalization to ensure that the classification performance was not derived from brain activity related to response selection but instead from neural activity related to perceptual processes. Therefore, we trained classifiers on the perceptual conditions and tested them on catch trials where the target was absent, but that were labeled according to the location of the participants' responses (8 possibilities). A second analysis was aimed to check whether superior posterior decoding probability for the 'Seen–Correct' correct trials rose from eye movements. To this aim, we fed the same classifier solely with EOG data, namely horizontal and vertical electro-oculograms.

## Acknowledgements

The authors express their gratitude to the UNICOG Consciousness team (Sebastien Marti, Aaron Schurger and Jaco Sitt) for their advice and insights and to NeuroSpin's support teams, particularly the LBIOM team for subject recruitment and the MEG team (Virginie van Wassenhove, Marco Buiatti, Leila Rogeaux) for data acquisition and analysis. Supported by INSERM, CEA, Collège de France, Fyssen grant to MS, DGA to JRK, an ERC grant 'NeuroConsc' to SD, Foundation Bettencourt-Schueller and the Roger de Spoelberch Foundation.

## Additional information

### Funding

| Funder | Grant reference | Author |
|--------|-----------------|--------|
| Fondation Fyssen | | Moti Salti |
| European Research Council (ERC) | NeuroConsc | Stanislas Dehaene |
| Direction Générale de l'Armement | | Jean-Remi King |

The funders had no role in study design, data collection and interpretation, or the decision to submit the work for publication.

### Author contributions

MS, Conception and design, Acquisition of data, Analysis and interpretation of data, Drafting or revising the article; SM, J-RK, Analysis and interpretation of data, Drafting or revising the article; LC, Acquisition of data, Drafting or revising the article; LP, SD, Conception and design, Analysis and interpretation of data, Drafting or revising the article

### Ethics

Human subjects: The study was approved by the by CPP IDF under the reference CPP 08 021. All subjects gave written informed consent and consent to publish before participating in the study.

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
