## [Decision Letter]

Thank you for sending your work entitled “Distinct cortical codes and temporal dynamics for conscious and unconscious percepts” for consideration at *eLife*. Your article has been favorably evaluated by Eve Marder (Senior editor), two reviewers, and Heidi Johansen-Berg, who is a member of our Board of Reviewing Editors.

The Reviewing editor and the reviewers discussed their comments before we reached this decision, and the Reviewing editor has assembled the following comments to help you prepare a revised submission.

All reviewers found the study well designed and carefully analysed. All reviewers felt the paper was a valuable addition to the literature and offered several advantages relative to prior work in this field. The following specific issues raised by reviewers should be addressed in a revision:

Essential points:

1) It would be useful to relate the decoding strategies used here to previous studies that used more traditional approaches to data analysis. For example, the authors could include a figure in which ERFs and ERPs are compared across the three main conditions (seen-correct, unseen-correct, unseen-incorrect). Specifically, it is worth reporting whether the difference between seen and unseen decoding accuracies observed in the late time period (<270 ms) corresponds to the N2-P3 waves reported in previous studies.

2) Unless we are missing something, the authors did not run any analyses in which multivariate decoding was used to classify seen vs. unseen responses. Of the many analyses reported, a difference in accuracy in decoding stimulus location was found for seen vs. unseen trials, but both were above chance. Was a classifier ever directly used to discriminate seen vs. unseen trials?

3) Perhaps the late activity involves online conscious control of the speeded hand-movement response rather than conscious perception? We recognize that the authors have included a control in which no stimulus is present and responses are made. However, that control does not fully address the potential concern. Would the authors predict similar late-stage decoding of stimulus location if a 500 ms stimulus was presented (so as to be obviously seen) and subjects were not required to make any responses? This requires additional discussion.

4) It is not clear from the Methods section how EEG and MEG data were separately analyzed or combined in the multivariate decoding. For example, Figure 2 shows “decoding location from MEG/EEG”, implying that signals from both were utilized in the decoding. More details regarding how these signals were separately analyzed or combined should be provided in the Methods section.

5) The authors should report reaction times for the speeded location identification response.

The authors may choose to comment on the following discussion points in a revision:

1) The authors make several well-informed predictions based on current theories or previous evidence, and the data appear to support these predictions. However, strong claims are often made based on very small effects. The authors should acknowledge this and should comment on the need for caution in interpreting a 1% difference in decoding accuracy between seen versus unseen trials. It would also be helpful to comment on why decoding accuracies were only just barely (though significantly) above chance to begin with.

2) If the late activity (270-800 ms) corresponds to conscious perception, the authors should consider explaining why the neural events underlying the conscious experience of a 33 ms line stimulus last 16 times longer (530 ms). Isn't it more intuitive that the conscious experience (and associated neural events) of a very brief stimulus would be correspondingly very brief?

3) Please provide more discussion of the finding that conscious representations seem to have a fixed duration of 160 ms which then gives rise to a new cycle, while unconscious representations decay slowly. If the authors could discuss this phenomena more extensively (e.g., do neural oscillations play a role? Does it reflect engagement of successive brain areas?), it would shed light on the mechanism behind this novel and interesting finding.

---

## [Author Response]

*1) It would be useful to relate the decoding strategies used here to previous studies that used more traditional approaches to data analysis. For example, the authors could include a figure in which ERFs and ERPs are compared across the three main conditions (seen-correct, unseen-correct, unseen-incorrect). Specifically, it is worth reporting whether the difference between seen and unseen decoding accuracies observed in the late time period (<270ms) corresponds to the N2-P3 waves reported in previous studies*.

The objective of our study was to go beyond existing data on the neural correlates of conscious perception, specifically by attempting to decode the neural dynamics of a specific content, both when it is consciously perceived and when it is processed non-consciously. This is the reason why we did not initially include the basic ERFs and ERPs for the three conditions. We agree, however, that this is important background information. Therefore, we have now included this information in the paper. The results indeed demonstrate classic differences in the N2 and P3 waves in ERPs, and the corresponding events in ERFs. We have added a specific subsection on this topic to the Results section (“Relation to previous findings from event-related responses”).

“Our fourth time window, where significantly improved location decoding was observed on seen compared to unseen trials […] the latter corresponds to a search for brain states encoding a specific consciously perceived content.”

*2) Unless we are missing something, the authors did not run any analyses in which multivariate decoding was used to classify seen vs. unseen responses. Of the many analyses reported, a difference in accuracy in decoding stimulus location was found for seen vs. unseen trials, but both were above chance. Was a classifier ever directly used to discriminate seen vs*. *unseen trials?*

Again, this was not the primary goal of our study. Numerous previous studies have studied the neural signatures of “Seen” and “Unseen” stimuli, but they were not able to identify the neural activity directly related to consciously perceived contents. Here we tried to pursue the neural dynamics underlying consciously perceived contents. Figuratively speaking, the old approaches have extracted the “canvas” on which conscious perception was painted while our approach is aimed at extracting the “painting” itself. We agree, however, that it is also useful to have additional information on the capacity to decode seen and unseen trials from M/EEG data. This information is now included in the paper (in the subsection headed “Relation to previous findings from event-related responses”):

”Training a classifier to use those event-related responses in order to separate “Seen-Correct” versus “Unseen-Correct” trials yielded a modest but significant classification success […]. This reduced the number of trials on which decoder was trained on and affected classification results.”

*3) Perhaps the late activity involves online conscious control of the speeded hand-movement response rather than conscious perception? We recognize that the authors have included a control in which no stimulus is present and responses are made. However, that control does not fully address the potential concern. Would the authors predict similar late-stage decoding of stimulus location if a 500ms stimulus was presented (so as to be obviously seen) and subjects were not required to make any responses? This requires additional discussion*.

Two valid points were raised here. The first point is the hypothesis that the late stage of the classification could reflect conscious response planning. We believe that our control, decoding target-absent trials, does address this concern, but we have added two arguments to support our claim that the decoding difference is related to conscious perception and not to conscious motor preparation (see subsection headed “Controls for hand and eye movements”). The revised paragraph now reads:

“Although it remains possible that conscious response planning modulates the late stages of location processing, several aspects of our experiments argue against a contribution of motor codes to our results. […] Nor can the improved classification performance for “Seen-Correct” over “Unseen-Correct”, since those trials involve exactly the same stimuli and responses.”

The second point concerns the generality of our results outside the specific experimental environment and task. In real-life situations, conscious contents are richer, continuous and do not necessarily require a report. The experimental design here used a very lean stimulus. Only one of its features (location) was of relevance to the participants, who had to explicitly report it and its visibility. Generalization beyond the experimental situation should consider whether the dynamics revealed here could be generalized to other features of the same stimuli, to other stimuli and to other tasks, including passive viewing. Here we set a methodological and theoretical framework and establish preliminary results. Obviously, a lot of additional work is needed to address these points of concern. We have added a final paragraph to the Discussion where we discuss the limits of our findings and argue, cautiously, that similar dynamics would be found if we had used clearly visible stimuli without any overt response. We base this conclusion on parallels with other experiments, for instance using the auditory local-global paradigm, where similar findings of a late activity correlated with conscious perception was observed even in the absence of any overt report. We do note, however, that there is a current debate in the literature concerning this very issue. A discussion of those points has been added in the final paragraph:

“While numerous previous studies have compared brain activity evoked by “seen” and “unseen” stimuli, the advantage of the present study is to specifically pursue the neural dynamics underlying a specific conscious content (stimulus location). […] With due caution, we therefore hypothesize that the neurophysiological correlates of conscious perception uncovered here might generalized to other experimental conditions.”

*4) It is not clear from the Methods section how EEG and MEG data were separately analyzed or combined in the multivariate decoding. For example,*
Figure 2
*shows “decoding location from MEG/EEG”, implying that signals from both were utilized in the decoding. More details regarding how these signals were separately analyzed or combined should be provided in the Methods section*.

All of our decoding was based on combined MEG and EEG signals. We have added a precise description of our methods in the subsection “Preprocessing” of the Materials and methods:

“…To combine data from MEG magnetometer and planar gradiometer as well as from EEG channels, a normalization procedure was first applied; the baseline standard deviation was estimated for each channel using all the trials in the experiment and then all the samples were divided by this std to yield a z-score which was entered in the classification algorithm”.

*5) The authors should report reaction times for the speeded location identification response*.

We reported the reaction times in the previous version in the Materials and methods section. We now moved it to the Results section:

“Mean localization RTs were 812.2±44 ms. Localization RTs for different critical conditions were as follow: 701.4±35 ms for “Seen-Correct” trials, 878.1±78 ms for “Unseen-Correct” trials and 1110±101 ms for “Unseen-incorrect” trials.”

*The authors may choose to comment on the following discussion points in a revision*:

*1) The authors make several well-informed predictions based on current theories or previous evidence, and the data appear to support these predictions. However, strong claims are often made based on very small effects. The authors should acknowledge this and should comment on the need for caution in interpreting a 1% difference in decoding accuracy between seen versus unseen trials. It would also be helpful to comment on why decoding accuracies were only just barely (though significantly) above chance to begin with*.

We agree and have toned down our claims whenever possible, particularly in an additional final paragraph of the Discussion, pointing to the limitations of our study. It is important to note that we are decoding single-trial responses. It is well-known that the signal-to-noise ratio in EEG and MEG is quite low, owing to a superimposition of ongoing spontaneous activity whose intensity can be more than ten times larger than evoked activity. Thus, the absolute decoding level is necessarily low. Our argument is not based on an absolute level of decoding (unlike, say, experiments that aim at efficient brain–computer interfaces), but on the mere presence of differentially decodable information, even if it remains at a low level.

*2) If the late activity (270-800ms) corresponds to conscious perception, the authors should consider explaining why the neural events underlying the conscious experience of a 33ms line stimulus last 16 times longer (530ms)*. *Isn't it more intuitive that the conscious experience (and associated neural events) of a very brief stimulus would be correspondingly very brief?*

We disagree with this intuition. Although the stimulus is brief, its internal representation may be long-lasting. There is considerable evidence that even a brief sensory event may cause an entire series of brain activations, lasting way beyond the duration of the stimulus. The vast majority of theories of consciousness, including Global Neuronal Workspace theory, acknowledge that there is a tight relation between conscious perception and working memory. Even briefly presented stimuli, if conscious, must remain efficiently coded and “meta-stable” for a duration long enough to allow reportability long after the stimulus is gone. Indeed, a direct relation between conscious perception and long-lasting metastability has been demonstrated in previous research. See for instance the following two papers:

King, J.-R., Gramfort, A., Schurger, A., Naccache, L., & Dehaene, S. (2014). Two distinct dynamic modes subtend the detection of unexpected sounds. PloS One, 9(1), e85791. http://doi.org/10.1371/journal.pone.0085791

Schurger, A., Sarigiannidis, I., Naccache, L., Sitt, J. D., & Dehaene, S. (2015). Cortical activity is more stable when sensory stimuli are consciously perceived. Proceedings of the National Academy of Sciences of the United States of America, 112(16), E2083–2092. http://doi.org/10.1073/pnas.1418730112

*3) Please provide more discussion of the finding that conscious representations seem to have a fixed duration of 160ms which then gives rise to a new cycle, while unconscious representations decay slowly. If the authors could discuss this phenomena more extensively (e.g., do neural oscillations play a role? Does it reflect engagement of successive brain areas?), it would shed light on the mechanism behind this novel and interesting finding*.

We agree that this rough duration of 160ms is a very interesting finding. We have added some discussion of its potential relation to theta frequencies, as the observed duration of successive conscious stages is compatible with a nearly rhythmic succession of processing stages at this frequency. We now cite several publications that have demonstrated a tight correlation between theta oscillations and conscious perception. A discussion of this point has been added to the Discussion:

“The observed characteristic peak-to-end duration of about 160ms is also compatible with the finding that conscious percepts tend to be locked to an ongoing theta (∼4–7 Hz) or high delta (∼1–3 Hz) rhythm […]. It thus seems plausible that conscious information is able to enter into a series of computational stages that are betrayed by a series of theta- or delta-like peaks (8).”